# Preliminary Evidence of the Differential Expression of Human Endogenous Retroviruses in Kawasaki Disease and SARS-CoV-2-Associated Multisystem Inflammatory Syndrome in Children

**DOI:** 10.3390/ijms242015086

**Published:** 2023-10-11

**Authors:** Emanuela Balestrieri, Elena Corinaldesi, Marianna Fabi, Chiara Cipriani, Martina Giudice, Allegra Conti, Antonella Minutolo, Vita Petrone, Marialaura Fanelli, Martino Tony Miele, Laura Andreozzi, Fiorentina Guida, Emanuele Filice, Matteo Meli, Sandro Grelli, Guido Rasi, Nicola Toschi, Francesco Torcetta, Claudia Matteucci, Marcello Lanari, Paola Sinibaldi-Vallebona

**Affiliations:** 1Department of Experimental Medicine, University of Rome Tor Vergata, 00133 Rome, Italy; chiara.cipriani@uniroma2.it (C.C.); giudicemartina94@gmail.com (M.G.); antonella.minutolo@uniroma2.it (A.M.); vita.petrone01@gmail.com (V.P.); fanellimarialaura@gmail.com (M.F.); miele@med.uniroma2.it (M.T.M.); grelli@med.uniroma2.it (S.G.); guidorasi@hotmail.com (G.R.); matteucci@med.uniroma2.it (C.M.); sinibaldi-vallebona@med.uniroma2.it (P.S.-V.); 2Pediatric Unit, Ramazzini Hospital, 41012 Carpi, Italy; e.corinaldesi@ausl.mo.it (E.C.); f.torcetta@ausl.mo.it (F.T.); 3Pediatric Emergency Unit, IRCCS Azienda Ospedaliero Universitaria di Bologna, 40126 Bologna, Italy; laurandreozzi@gmail.com (L.A.); flo.guida@gmail.com (F.G.); emanfilice@gmail.com (E.F.); matte.meli92@gmail.com (M.M.); marcello.lanari@unibo.it (M.L.); 4Department of Biomedicine and Prevention, University of Rome Tor Vergata, 00133 Rome, Italy; conti.allegra@gmail.com (A.C.); toschi@med.uniroma2.it (N.T.); 5Martinos Center for Biomedical Imaging and Harvard Medical School, Boston, MA 02129, USA; 6National Research Council, Institute of Translational Pharmacology, 00133 Rome, Italy

**Keywords:** human endogenous retroviruses, HERVs, inflammation, COVID-19, Kawasaki disease, multisystem inflammatory syndrome in children

## Abstract

Multisystem inflammatory syndrome in children (MIS-C) is a postinfectious sequela of severe acute respiratory syndrome coronavirus 2 (SARS-CoV-2), with some clinical features overlapping with Kawasaki disease (KD). Our research group and others have highlighted that the spike protein of SARS-CoV-2 can trigger the activation of human endogenous retroviruses (HERVs), which in turn induces inflammatory and immune reactions, suggesting HERVs as contributing factors in COVID-19 immunopathology. With the aim to identify new factors involved in the processes underlying KD and MIS-C, we analysed the transcriptional levels of HERVs, HERV-related genes, and immune mediators in children during the acute and subacute phases compared with COVID-19 paediatric patients and healthy controls. The results showed higher levels of HERV-W, HERV-K, Syn-1, and ASCT-1/2 in KD, MIS-C, and COV patients, while higher levels of Syn-2 and MFSD2A were found only in MIS-C patients. Moreover, KD and MIS-C shared the dysregulation of several inflammatory and regulatory cytokines. Interestingly, in MIS-C patients, negative correlations have been found between HERV-W and IL-10 and between Syn-2 and IL-10, while positive correlations have been found between HERV-K and IL-10. In addition, HERV-W expression positively correlated with the C-reactive protein. This pilot study supports the role of HERVs in inflammatory diseases, suggesting their interplay with the immune system in this setting. The elevated expression of Syn-2 and MFSD2A seems to be a distinctive trait of MIS-C patients, allowing to distinguish them from KD ones. The understanding of pathological mechanisms can lead to the best available treatment for these two diseases, limiting complications and serious outcomes.

## 1. Introduction

Kawasaki disease (KD) is a febrile systemic vasculitis of the small- and medium-sized arteries that usually affects children younger than 5 years of age [1]. KD represents the leading cause of acquired heart disease in childhood because of its potential involvement of coronary arteries with dilations and aneurysms, and subsequent myocardial ischemia and sudden death. The incidence varies a lot all over the world, with the highest incidence in the countries of East Asia. In Italy, particularly in Emilia-Romagna, the region where Bologna and Carpi are located, the incidence was reported to be 16.4 cases/100,000 children under 5 years [2]. The exact pathological mechanisms are still unknown: probably an environmental factor, such as an infective agent, triggers the excessive inflammatory response in genetically predisposed children. However, to date, no infectious agent has been proven to be consistently associated with KD [3,4,5,6]. The recent human coronavirus disease 2019 (COVID-19), caused by the novel severe acute respiratory syndrome coronavirus 2 (SARS-CoV-2) infection, has shown different patterns of disease progression. SARS-CoV-2-positive children are usually asymptomatic or develop mild disease and have a better prognosis than their adult counterparts [7,8]. Nevertheless, SARS-CoV-2-positive children are at risk of developing a postinfectious complication called multisystem inflammatory syndrome in children (MIS-C), characterised by severe systemic inflammation affecting multiple organs and tissues, such as the abdominal, respiratory, and cardiovascular systems [9]. Cardiac dysfunction and shock can rapidly progress, leading to the need of intensive care unit admission. MIS-C clinical features overlap with KD, and therefore MIS-C has been also called Kawasaki-like syndrome [10,11,12]. Its aetiology is not fully known [13]: the spectrum is presumed to be the result of a cytokine storm leading to a multiorgan inflammation [14,15] triggered by a viral agent in genetically predisposed children [16,17]. Recently, our research group has highlighted the implication of the human endogenous retrovirus-W (HERV-W) in COVID-19. Indeed, high expression levels of HERV-W ENV protein have been shown i in the blood cells of COVID-19 patients, which is associated with several clinical and biological parameters linked to patient status at hospitalisation and disease progression, suggesting HERVs as contributing factors in COVID-19 immunopathology [18]. Furthermore, the expression levels of different HERVs change according to the severity of MIS-C, showing an increase in children with a mild/moderate form and a decrease in those with severe MIS-C and COVID-19 compared to healthy controls [19]. HERVs are genetic elements that originate from ancestral infection and proliferation within the germ-line cells by exogenous retroviruses [20]. Since HERVs have been integrated as proviruses in germ cell lineages, they have been vertically transmitted from ancestors to descendants, and currently they make up about 8% of the human genome [21]. Interindividual variations, such as copy number variations, unfixed copies, and polymorphisms, have been widely described [22]. As HERVs make up a considerable part of our genome and their expression has been found in several tissues and during evolution, they have been co-opted for physiological functions, especially during pregnancy [23,24,25,26]. Moreover, in terms of the inflammatory process and HERVs, there seems to be a mutual influence, since HERV-derived products can be sensed by distinct types of pattern-recognition receptors (PRRs), stimulating the innate immune response [27]. As such, the HERV-W family interacts with TLRs, inducing a prominent proinflammatory response characterised by the release of several cytokines, such as IL-1β, IL-6, and TNF-α [28]. On the other side, the inflammatory effectors induced by HERVs could, in turn, further increase HERV activity [29,30]. Indeed, transcription factors can bind the LTR sites of HERV-K, especially those associated with NF-kB, and also the interferon (IFN)-stimulated regulatory element, which increases the expression of proinflammatory cytokines, including type I IFNs [29,31]. TNF-α was able to increase the RNA expression of HERV-H, HERV-K, and HERV-W, driving the translocation of NF-kB into the nucleus with subsequent binding to sites presenting HERV LTRs [32]. Although HERVs are generally silenced, peculiar HERV copies can be activated by different environmental stimuli, leading to the expression of immunopathogenic proteins. However, their relationship as direct causative agents or indirect contributors of the disease is not fully elucidated [33,34,35]. 

With the aim to identify new factors that may contribute to the immunopathogenic processes underlying KD and MIS-C, here we analysed the transcriptional levels of HERVs, putative HERV receptors, and immune mediators in affected children during acute and subacute phases in comparison with COVID-19 paediatric patients and healthy controls, and we also evaluated any possible correlation among HERV levels and different laboratory parameters recorded.

## 2. Results

### 2.1. Clinical and Laboratory Characteristics of Kawasaki Disease and Multisystem Inflammatory Syndrome in Children Patients

We analysed the clinical and laboratory information of patients with KD and patients with MIS-C during the acute and the subacute phase. The COV patients and HCs were best-matched for age and sex with the study cohort (demographic characteristics are detailed in Table 1). Clinical and laboratory characteristics of KD and MIS-C are shown in Table 2: all children were febrile. COV patients presented respiratory symptoms (8/10) and fever (8/10).

### 2.2. HERV ENV, Syncytins, and Their Putative Receptors Are Differentially Expressed in Blood Samples from Paediatric Patients with Kawasaki Disease and Multisystem Inflammatory Syndrome in Children

The transcriptional activity was investigated in blood samples from children during the acute and subacute phase of Kawasaki disease (KDa and KDsa, respectively) and multisystem inflammatory syndrome (MIS-Ca and MIS-Csa, respectively), and the values were compared with those of paediatric COVID-19 patients (COV) during the acute phase and healthy controls (HCs) by quantitative RT real-time PCR analysis. Results are represented as box plots in Figure 1 (panels A–C), while *p*-values resulting from the Mann–Whitney test are detailed in Figure 1 (panel D). Median values and interquartile ranges are reported in Appendix A. When comparing with HCs, a general increase in the mRNA of all the analysed genes in patients affected by KD, MIS-C, and COVID-19 was found. In particular, higher levels of HERV-W, HERV-K, Syn-1, and ASCT-1 and 2 were found in blood samples from KD, MIS-C, and COV patients, while higher levels of Syn-2 and MFSD2A were found only in MIS-C patients, regardless of the disease phase. Taking into account that MIS-C develops following a SARS-CoV-2 infection, we also compared them to COV, detecting higher levels of HERV-W, Syn-1, ASCT-1 and 2, Syn-2, and MFSD2A in the former group. When comparing KDa and MIS-Ca, the mRNA levels of HERV-W, HERV-K, Syn-1, Syn-2, ASCT-1, and MFSD2A were higher in MIS-Ca and higher levels of HERV-K, Syn-2, and MFSD2A were maintained in MIS-Csa with respect to KDsa. Comparing phases of the disease, higher levels of HERV-K were detected in KDsa patients respect to KDa patients, while higher levels of HERV-W, HERV-K, Syn-1, and ASCT-1 were found in MIS-Ca compared to MIS-Csa. All the statistically significant differences observed were associated with a large effect size (Cohen’s d ≥ 0.8). 

### 2.3. Kawasaki Disease and Multisystem Inflammatory Syndrome in Children Share the Deregulation of Inflammatory and Regulatory Cytokines, but They Differ in the Transcriptional Profile of Toll-like Receptors

In the same samples described above, the mRNA levels of cytokines (IL-1β, IL-6, IL-10, TNF-α, MCP-1), interferon-gamma (IFN-γ), and Toll-like receptors (TLR-3, TLR-4, TLR-9) were also assessed by quantitative RT real-time PCR analysis. Results are represented as box plots in Figure 2 (panels A–C), and *p*-values resulting from the Mann–Whitney test are detailed in Figure 2 (panel D). Median values and interquartile ranges are reported in Appendix A. All the three diseases are characterised by a statistically significant increase in the mRNA levels of the cytokines when compared to HCs. For all TLRs analysed, statistically significant differences were found only in COV and MIS-Ca and MIS-Csa patients compared to HCs. Comparing SARS-related diseases, higher levels of IL-6, IL-10, MCP-1, TLR-3, and TLR-9 were found in COV patients, while TNF-α levels were found to be higher in MIS-Ca. When comparing KDa and MIS-Ca patients, only the expression levels of TNF-α were higher in the latter, while IL-10 levels were higher in the former group. Higher levels of IL-10 and MCP-1 were found in MIS-Csa with respect to KDsa. Considering the progression of the disease, in KD, higher levels of IL-10, TNF-α, and MCP-1 were found during the acute phase compared with the subacute phase. Concerning MIS-C, higher levels of TNF-α, TLR-3, TLR-4, TLR-7, and TLR-9 were found during the acute phase with respect to the subacute phase, while IL-6 and IL-10 levels were found to be higher in the subacute phase. All statistically significant differences were associated with a moderate to large effect size (Cohen’s d ≥ 0.5).

### 2.4. The Correlation Analysis Demonstrates an Association among HERVs and Cytokine Expression Only in Patients with Kawasaki Disease and Multisystem Inflammatory Syndrome

Pairwise associations between continuous variables were tested through the Spearman correlation coefficient, and statically significant results are reported in Figure 3. The analysis revealed positive correlations between HERV-W and IL-6 (Rho = 0.667, *p* = 0.05) and HERV-K and IL-10 (Rho = 0.833, *p* = 0.005) in COV patients. In MIS-Ca, negative correlations were found between HERV-W and IL-10 (Rho = −0.818, *p* = 0.004) and between Syn-2 and IL-10 (Rho = −0.839, *p* = 0.002), while positive correlations were found between HERV-K and IL-10 (Rho = 0.685, *p* = 0.029). In addition, HERV-W mRNA levels positively correlated with CRP (Rho = 0.788, *p* = 0.014). No correlations were found in KD patients and HCs.

## 3. Discussion

Our findings point out a general deregulation of the transcriptional activity of HERVs, syncytin-1, and its putative receptors, as well as inflammatory mediators in patients affected by COVID-19, MIS-C, and KD. In COVID-19 patients, the potential contribution of HERVs in the etiopathogenesis has been extensively suggested, demonstrating their activation in different clinical settings. Specifically, the HERV-W envelope (ENV) mRNA and protein were found to be highly expressed in the blood cells and associated with disease severity and pulmonary involvement. Moreover, ENV protein expression in lymphocytes reflected the respiratory outcome during hospitalisation [18]. Furthermore, in tracheal aspirates from COVID-19 patients under intermittent mandatory ventilation, the expression of HERV-K was found at a high level [36]. More recently, we demonstrated the activation of HERVs and mediators of the innate immune response in the initial site and at the early stage of SARS-CoV-2 infection closely related to disease severity, sustaining this complex profile as a predictive biomarker of COVID-19 severity and patient outcome [37]. Interestingly, here we also report that paediatric COV showed an altered HERV transcriptional profile, in line with a previous report [19]. Notably, only patients with MIS-C, in addition to having high values of HERVs, Syn-1, and its putative receptors, showed elevated mRNA levels of Syn-2 and its receptor named MFSD2A. Syncytins represent the most striking example of how HERVs in the course of coevolution with humans have been co-opted to serve beneficial functions by being involved in the successful placentation and in the acquisition of the maternal immune tolerance against the foetus [38,39]. The different expression of syncytins in the two SARS-related diseases may be attributable to other functions than the well-described fusogenic properties. Actually, Syn-1 may also regulate the production of inflammatory mediators, and Syn-2 differs from Syn-1 in having an immunosuppressive domain involved instead of preventing the activation of the immune response by the mother to foetal alloantigens [40,41]. The observation of an increased transcriptional activity of Syn-2 and MFSD2A only in peripheral blood from MIS-C patients is intriguingly and potentially linked to an effort to limit the massive inflammatory response characteristic of the disease. Important to note, this expression profile is also maintained in MIS-C patients during the subacute phase towards the resolution of the disease. Conversely, a decrease in HERVs and Syn-1 and its receptors was found in patients during the MIS-C subacute phase, likely attributable to the reduction of the infectious stimulus.

Here, we show for the first time that HERVs, Syn-1, and its putative receptors are also highly expressed in KD patients, although the observed levels do not reach those observed in MIS-C patients. Given the inflammatory nature of KD and the ability of HERVs to modulate the inflammatory response, it is possible to speculate that HERVs may contribute to the immunopathogenesis of KD as well. Of note, the expression levels found in KD patients did not reach those observed in COV and MIS-C patients. Taking into account the phases of the KD, an increased transcriptional activity of HERV-K was found in the subacute phase with respect to the acute one. This observation is supported by a recent view of the physiological function of HERV-K, able to activate the interferon secretion in COVID-19 patients, favouring the containment and the resolution of the infection [42]. Based on these findings, altered levels of HERVs and Syn-1 appear to be a common molecular trait in SARS-related diseases and KD. In contrast, Syn-2 and MFSD2A could be suggested as hallmarks of MIS-C, able to characterise the disorder from others with overlapping symptomatology.

In the same patients, we also analysed several inflammatory and regulatory cytokines, IFN-γ, and TLRs, demonstrating that MIS-C and KD patients shared the deregulation of all these markers, except for TLRs. Indeed, comparing KD and MIS-C patients during the acute phase, only the expression levels of TNF-α were higher in the latter, while the IL-10 levels were higher in the former group, and during the subacute phase, higher levels of IL-10 and MCP-1 were found in MIS-C with respect to KD. Interestingly, focusing on the progression of the diseases, in KD, higher levels of IL-10, TNF-α, and MCP-1 were found during the acute phase compared with the subacute phase, while in MIS-C, higher levels of TNF-α, TLR-3, TLR-4, TLR-7, and TLR-9 were found during the acute phase with respect to the subacute phase, while IL-6 and IL-10 levels were found to be higher in the subacute phase. Our findings are in line with a huge amount of literature describing the “cytokine storm” as key mediator of damage in COVID-19, and that many proinflammatory cytokines were increased both in MIS-C and KD patients [9,43]. In KD patients have been reported the upregulation of most TLR (1, 4, 5, and 8) expressions compared to both the healthy and febrile controls, likely due a hypomethylated *status* [44]. Our results are only partly in line with these observations. As such, no significant differences were found in our KD cohort with respect to healthy controls for TLR-4, and this could be due to the relatively small sample size analysed. Concerning COVID-19 and MIS-C patients, a general increase in all TLRs evaluated has been found, and this was in agreement with the well-known functions of viral elements and damage associated to host molecules able to act as TLR ligands in these diseases [45]. Patients with COVID-19 show a complex immunoinflammatory response as a result of the host–virus interaction [46], and in severe patients, high serum levels of proinflammatory cytokines and chemokines, including IL-2, IL-6, IL-1β, IL-8, IL-17, G-CSF, GM-CSF, IP10, MCP1, MIP1α, and TNF-α, leading to host cell death, and organ injury was also observed [47,48,49]. Similarly, in MIS-C patients, the release of inflammatory mediators is induced by the activation of TLRs expressed, resulting in NF-kB activation [50]. Our results highlighted that the increased mRNA levels of TLRs in COV paediatric and MIS-C patients parallels with HERV activation, and this could be due the ability of HERVs to activate PRRs, evoking the production of proinflammatory mediators [51]. In fact, the HERV-W recombinant ENV protein, though TLR-4 receptor activation, leads to the production of proinflammatory cytokines both in human and murine monocytes [52]. The Spearman analysis also demonstrated different correlations among HERVs and cytokine mRNA levels only in MIS-C and COV paediatric patients. Specifically, HERV-W positively correlated with IL-6 expression and HERV-K with IL-10 in COV patients. In MIS-C patients in the acute phase, a negative correlation was found between HERV-W and IL-10 expression and between Syn-2 and IL-10, while between HERV-K and IL-10, a positive correlation was observed. The positive association between HERV-W with IL-6 expression and the negative association with IL-10 are in line with the immunopathogenic role of the ENV HERV-W protein as described in MS and recently in COVID-19 [18,52,53]. Indeed, in MS, the ENV protein interacts with TLR4, leading to the induction of a sustained proinflammatory response, which includes the release of numerous cytokines, such as IL-1β, IL-6, and TNF-α [54]. Intriguingly, a temporal correlation between inflammatory markers and HERV expression was already established using an ex vivo healthy donor PBMC stimulation approach, revealing that the induction of ENV expression by the SARS-CoV-2 spike protein occurs prior to the expression of IL-6 [18]. In COV and MIS-C patients, we also found a positive correlation between HERV-K and IL-10, in line with a previous study showing that HERV-K induced IL-10 release from human PBMCs, suggesting that the immunosuppressive activity may be an intrinsic property of some HERVs [55]. The interplay between HERVs and inflammatory mediators is attributable to the fact that HERV antigens can be recognised by the innate immune system as “self-determinants”, but also as potential pathogens able to evoke the production of proinflammatory mediators, which in turn can activate and trigger the adaptive immune response, sustaining different degrees of chronic inflammation [51]. Furthermore, a reciprocal influence between the inflammatory process and HERVs, by removing the blocks limiting the expression and regulation of several HERV-mediated genes, causes a new imbalance in gene expression, which favours an increased instability and, in turn, aggravates the inflammatory condition [29], as indicated in different complex disorders characterised by an inflammatory landscape, including autoimmune diseases [56], type 1 diabetes [57], and neurological and psychiatric disorders [33,34]. In addition, only in MIS-C during the acute phase HERV-W mRNA levels positively correlate with CRP. This observation is in line with the correlation between HERV-W GAG or ENV antigenemia and CRP levels described in schizophrenic patients, corroborating the hypothesis that HERV-W sustains a chronic inflammatory state [58].

The main limitations of the study are the small sample size and the difference in terms of the male/female ratio; thus, our work, should be considered as a pilot study supporting the role of HERVs in inflammatory diseases. Notwithstanding, in this complex scenario, HERVs and their interaction with the immune system could represent a new piece in the puzzle of the immunopathogenic process underlying KD and MIS-C, for which a single determinant alone could not be sufficient to explain their multifaceted nature. Moreover, as the symptoms and course of MIS-C may resemble those of KD, the diagnosis of these syndromes can be challenging, and the distinction between them should require additional clinical attention. Further efforts are needed to candidate HERVs together with inflammatory mediators as complex biomarkers, including the chance to move toward protein-focused studies. Notably, the elevated mRNA levels of Syn-2 and MFSD2A seems to be a distinctive trait of MIS-C patients at the transcriptional level, allowing us to distinguish them from KD patients. The early diagnosis and the understanding of the pathological mechanisms can lead to the best available treatment for these two diseases, limiting complications and more serious outcomes.

## 4. Materials and Methods

### 4.1. Study Design and Participants

We conducted a multicentre prospective study including children diagnosed with KD (*n* = 8) and MIS-C (*n* = 17) between October 2020 and June 2021 in 2 paediatric units (IRCCS- St. Orsola University Hospital, Bologna, and Ramazzini Hospital, Carpi) in Emilia-Romagna (Italy), and children diagnosed with acute COVID-19 (COV) (*n* = 10) admitted at the IRCCS-St. Orsola Hospital Pediatric Emergency Department in the same time interval. All KD diagnoses and treatments were made in accordance with the 2017 American Heart Association (AHA) guidelines [1]. MIS-C diagnosis was made according to WHO criteria [59]. For both KD and MIS-C, the onset of illness was defined as the first day of fever. The time interval between the disease onset and the 10th day of fever was defined as the “acute phase”, while the one between the 11th and the 20th day after the fever onset was defined as the “subacute phase”. Considering the responsiveness to standard treatment with intravenous immunoglobulin (IVIG), patients were divided into responders and nonresponders. We previously described the definition of IVIG responsiveness, echocardiographic timing, and findings and their evolution. Briefly, nonresponse to IVIG was defined when fever was persistent or recrudescent 36 h after the end of IVIG infusion. An echocardiography was performed to all KD and MIS-C patients before the 10th day from fever onset (acute phase) and between the 11th and 20th day (subacute phase). Coronary lesions (CALs) were classified following the AHA KD guidelines [1] as normal, ectasia, and aneurysm.

COV children were diagnosed those patients who tested positive for SARS-CoV-2 RT-PCR in swab samples and reporting symptoms compatible with COVID-19. The onset of illness was defined as the day when the first symptom or sign occurred. 

Age- and sex-matched healthy controls (HCs) (*n* = 19) were recruited among children attending the outpatient facilities of the “Tor Vergata” Hospital (Rome, Italy) for routine examinations, used in our previous studies [60,61]. None of them reported neurological or psychiatric disorders or the presence of ongoing infections in their medical history. All individuals enrolled as controls were not taking any medications at the sampling time. 

For all individuals included in the study, a dedicated blood sample was collected in EDTA tubes and stored at −80 °C.

### 4.2. Data Collection

In order to collect data regarding KD, MIS-C, and COV patients, a database was prospectively created and subsequently retrospectively reviewed. Laboratory data of KD and MIS-C patients were collected during the acute and subacute phases, while COV children were tested only during the acute stage of infection.

### 4.3. Ethical Approval

The study was conducted according to the guidelines of the Declaration of Helsinki. For patients’ and controls’ enrolment and data collection, the study was approved by the local Ethics Committee (Comitato Etico Area Vasta Emilia Centro—AVEC, Bologna, Italy; protocol codes: no. 78/2021/Sper/AOUBo) and by the University Hospital of “Tor Vergata” Ethics Committee (no. 77/13), respectively. Written informed consent was obtained from the parents.

### 4.4. Sample Preparation, RNA Extraction, and Quantitative RT Real-Time PCR

Total RNA isolation from whole-blood samples was performed as previously described in [18,60]. Concentration of RNA was evaluated by Nanodrop DS 11 (DeNovix), and samples analysed showed a ratio of 260/280 of about 2.0 and a concentration ranging from 50 ng/mL to 250 ng/mL. DNase-treated RNA (100 ng) was reverse-transcribed into cDNA as previously described [18]. The quantitative evaluation of the transcriptional levels of (i) the env gene of HERV-W ENV, HERV-K; (ii) syncytin-1 (Syn-1) and syncytin-2 (Syn-2); (iii) HERV-related genes: alanine/serine/cysteine/threonine-preferring transporter 1 and 2 (ASCT-1, ASCT-2), the major facilitator superfamily-domain-containing 2A (MFSD2A); (iv) inflammatory and regulatory cytokines: interleukin-1β (IL-1β), IL-6, IL-10, tumour necrosis factors-α (TNF-α), interferon (INF-γ), Toll-like receptors (TLR-3, TLR-4, TLR-9), and chemokines such as monocyte chemoattractant protein 1 (MCP-1) was performed as previously described [18] with specific primer pairs, as listed in Table 3.

To set up the real-time reaction, a serial dilution (10-fold) was done to calculate the efficiencies and correlation coefficient by the formula [efficiency = 10 (−1/slope)], and all primer pairs used showed an efficiency ranging from 0.96 to 0.97. On the base of the set-up results, primer pairs were added at specific concentration, as follows: HERV-W ENV, ASCT-1, MFSD2A (100 nM), IL-1β, IL-6, IL-10, TNF-α, INF-γ, MCP-1 (150 nM), HERV-K, Syn-1, Syn-2, ASCT-2, TLR-3, TLR-4, and TLR-9 (200 nM). Each sample was analysed in triplicate, and a negative control (no template reaction) was included to check for any possible contamination. The housekeeping gene glyceraldehyde-3-phosphate dehydrogenase (GAPDH) was used to normalise the results, the quantification was performed as previously described [18], and the relative expression was calculated as follows:2 ^− [ΔCt (sample) − ΔCt (calibrator) = 2 − ΔΔCt^(1)
where ΔCt (sample) = [Ct (target gene) − Ct (housekeeping gene)] and ΔCt (calibrator) was the mean of ΔCT of all the samples from healthy control children.

### 4.5. Statistical Analysis 

Categorical variables were summarised as absolute and relative frequencies. Numerical variables were reported as medians and interquartile ranges (IQRs). The Mann–Whitney U-test was used to compare HERVs, HERV-related sequences, their putative receptors, cytokines, interferons, and Toll-like receptor expression levels in PBMCs from paediatric patients with KD and MIS-C, and COV patients and HCs. The comparison between clinical parameters collected from KD and MIS-C patients during the acute and subacute phases were also analysed by the Mann–Whitney U-test. Cohen’s d was estimated to measure the sizes of associations between each pair of variables. Effect sizes were classified as small (d = 0.2), medium (d = 0.5), and large (d ≥ 0.8) [68]. Pairwise associations between continuous variables were tested through the Spearman correlation coefficient. Statistical analyses were carried out using SPSS software, and statistically significant comparisons were considered when *p* < 0.050.

## Figures and Tables

**Figure 1 ijms-24-15086-f001:**
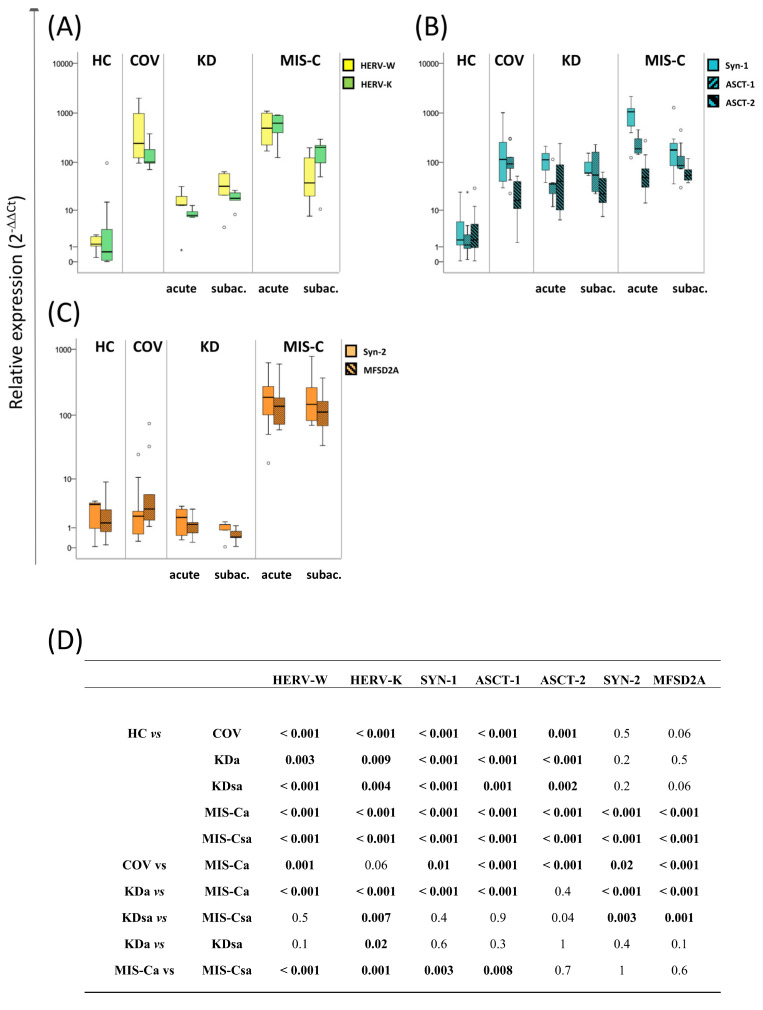
The mRNA levels of the ENV gene of HERVs, HERV-related sequences, and their putative receptors in blood samples from paediatric patients with Kawasaki disease, COVID-19, multisystem inflammatory syndrome, and healthy controls. The expression levels of (**A**) the env gene of HERV-W and HERV-K, (**B**) syncytin-1 (Syn-1), alanine/serine/cysteine/threonine-preferring transporter 1 and 2 (ASCT-1, ASCT-2), and (**C**) syncytin-2 (Syn-2), and the major facilitator superfamily-domain-containing 2A (MFSD2A) were analysed by qRT-PCR in blood samples from children during the acute and subacute phase of Kawasaki disease (KDa and KDsa), multisystem inflammatory syndrome (MIS-Ca and MIS-Csa), paediatric COVID-19 patients (COV) during the acute phase, and healthy controls (HCs). The results are represented as box plots depicting mild (black dot) and extreme outliers (asterisk) for each group. (**D**) The *p*-values for groupwise differences examined by the nonparametric Mann–Whitney U-test (statistical significance is defined when *p* < 0.050).

**Figure 2 ijms-24-15086-f002:**
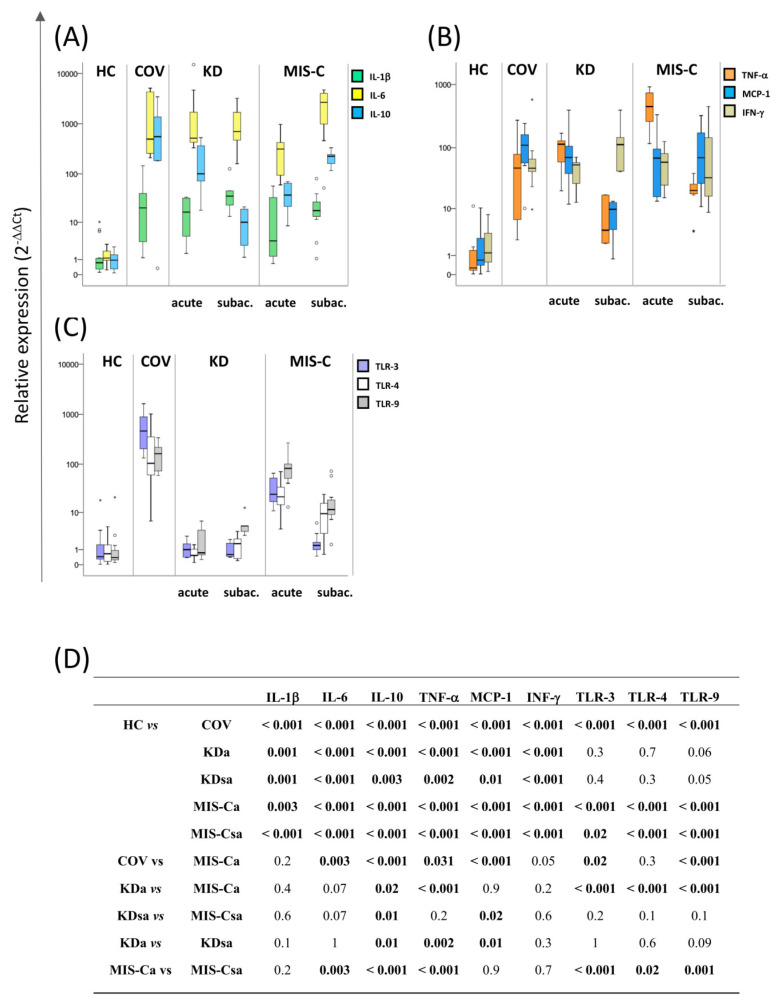
The mRNA levels of inflammatory and regulatory cytokines and interferon-gamma and Toll-like receptors in blood samples from paediatric patients with Kawasaki disease, COVID-19, multisystem inflammatory syndrome, and healthy controls. The mRNA levels of (**A**) inflammatory and regulatory cytokines: interleukin-1β (IL-1β), IL-6, and IL-10, (**B**) tumour necrosis factor-α (TNF-α), interferon (INF-γ), and monocyte chemoattractant protein 1 (MCP-1), and (**C**) Toll-like receptors (TLR-3, TLR-4, and TLR-9) were analysed by qRT-PCR in blood samples from children during the acute and subacute phase of Kawasaki disease (KDa and KDsa), multisystem inflammatory syndrome (MIS-Ca and MIS-Csa), paediatric COVID-19 patients (COV) during the acute phase, and healthy controls (HCs). The results are represented as box plots depicting mild (black dot) and extreme outliers (asterisk) for each group. (**D**) The *p*-values for groupwise differences examined by the nonparametric Mann–Whitney U-test (statistical significance is defined when *p* < 0.050).

**Figure 3 ijms-24-15086-f003:**
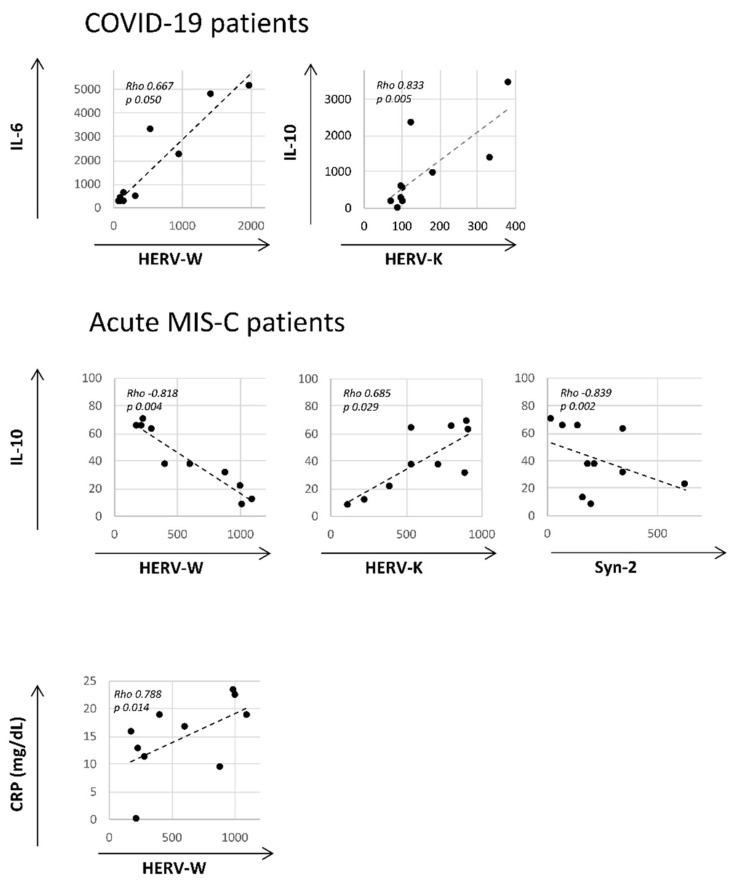
Scatter plots of HERVs expression levels (X-axis) versus cytokines and CRP levels (Y-axis) in MIS-Ca and COV patients. Pairwise associations between continuous variables were tested through the Spearman correlation test. The Rho coefficients and *p*-values were reported.

**Table 1 ijms-24-15086-t001:** Demographic characteristics of the cohort study.

	KD	MIS-C	COV	HCs
Number of children	8	17	10	19
Male n.	2	11	8	11
Male/Female ratio	0.25	0.65	0.8	0.58
Median age (IQR) *	110 (17–152)	68 (47–105)	33.5 (12.5–80.75)	48 (16–96)

Legend: KD stands for Kawasaki disease; MIS-C stands for multisystem inflammatory syndrome; COV stands for acute COVID-19; HCs stands for healthy controls; * months.

**Table 2 ijms-24-15086-t002:** Clinical presentation and laboratory tests of the acute and subacute stages of KD and MIS-C patients.

Clinical Manifestations		
	KD (n = 8)	MIS-C (n = 17)
Respiratory symptoms, n (%)	1 (12.5%)	4 (23.5%)
Conjunctival hyperaemia, n (%)	8 (100%)	12 (70.6%)
Extremity changes, n (%)	3 (37.5%)	6 (35.3%)
Skin rash, n (%)	8 (100.0%)	10 (58.8%)
Oral changes, n (%)	5 (62.5%)	9 (52.9%)
Cervical lymphadenopathy, n (%)	6 (75.0%)	3 (17.6%)
Abdominal involvement, n (%)	5 (62.5%)	14 (82.4%)
Hypotension, n (%)	0 (0%)	9 (64.7%)
Day of standard treatment, median (IQR)	6.5 (4.7–8.00)	5 (4.0–6.25)
Nonresponders, n (%)	2 (25.0%)	3 (17.6%)
Coronary artery involvement		
*Acute phase*		
CALs, n (%)	1 (12.5%)	5 (29.4%)
*Subacute phase*		
CALs, n (%)	1 (12.5%)	2 (11.8%)
Laboratory values		
*Acute phase*		
WBC × 10^9^/L, median (IQR)	14.6 (13.8–16.4)	9.1 (5.8–14.7)
N%, median (IQR)	71.4 (69.9–79.2)	78.0 (74.1–84.5)
L%, median (IQR)	18.9 (12.8–22.1)	13.0 (10.8–20.3)
E%, median (IQR)	0.7 (0.5–1.4)	0.3 (0.1–2.0)
NLR, median (IQR)	4.1 (3.2–6.3)	5.8 (3.7–7.6)
RBC × 10^12^/L, median (IQR)	4.2 (4.1–4.8)	4.0 (3.9–4.3)
Hb g/dL, median (IQR)	11.8 (11.0–11.9)	10.9 (10.5–11.6)
PLT × 10^9^/L, median (IQR)	344 (329–397)	161 (129–289)
CRP mg/dL, median (IQR)	11.5 (6.3–16.4)	17.7 (12.8–23.1)
Albumin g/dL, median (IQR)	3.7 (3.5–4.0)	3.1 (3.0–3.8)
Sodium mmol/L, median (IQR)	135 (133–138)	134 (131–136)
ALT IU/L, median (IQR)	24 (19–162)	29 (21–50)
Troponin ng/mL, median (IQR)	5.7 (4.0–7.75)	84.2 (11.3–124.1)
BNP ng/L, median (IQR)	139.0 (132.0–154.0)	481.5 (238.3–1437.3)
*Subacute phase*		
WBC × 10^9^/L, median (IQR)	12.2 (9.8–14.8)	10.7 (12.6–15.7)
N%, median (IQR)	36.7 (26.1–50.3)	62.6 (53.0–75.4)
L%, median (IQR)	47.5 (36.2- 60.1)	24.7 (19.1–36.6)
E%, median (IQR)	2.9 (0.9–5.2)	0.2 (0.1–0.5)
NLR, median (IQR)	0.9 (0.5–1.1)	2.4 (1.5–4.0)
RBC × 10^12^/L, median (IQR)	3.9 (3.7–4.4)	4.2 (3.8–4.3)
Hb g/dL, median (IQR)	10.9 (9.9–12.2)	11.2 (10.2–11.8)
PLT × 10^9^/L, median (IQR)	649 (479–790)	419 (285–550)
CRP mg/dL, median (IQR)	0.7 (0.3–1.1)	1.4 (0.5–1.7)
Albumin g/dL, median (IQR)	3.4 (3.0–3.6)	3.9 (3.5–4.3)
Sodium mmol/L, median (IQR)	139 (137–140)	139 (137–141)
ALT IU/L, median (IQR)	22 (17–42)	43 (25–60)

Legend: KD stands for Kawasaki disease; MIS-C stands for multisystem inflammatory syndrome in children; COV stands for COVID-19; IVIG stands for intravenous immunoglobulin; WBC stands for white blood cells; N% stands for neutrophil percentage values; L% stands for lymphocyte percentage values; E% stands for eosinophil percentage values; NLR stands for neutrophils–lymphocytes ratio; RBC stands for red blood cells count; Hb stands for haemoglobin; PLT stands for platelets; CRP stands for C-reactive protein; ALT stands for alanine aminotransferase; BNP stands for brain natriuretic peptide.

**Table 3 ijms-24-15086-t003:** Primer pairs used in qRT real-time PCR analysis.

Gene		Forward Primers (5′→3′)	Reverse Primers (5′→3′)	Ref.
HERV-W	AF331500	GTATGTCTGATGGGGGTGGAG	CTAGTCCTTTGTAGGGGCTAGAG	[62]
HERV-K	AF1646	GCCATCCACCAAGAAAGCA	AACTGCGTCAGCTCTTTAGTTGT	[60]
Syn-1	NM_001130925.2	ACTTTGTCTCTTCCAGAATCG	GCGGTAGATCTTAGTCTTGG	[63]
ASCT-1	NM_001193493.2	CTGGTGTTAGGAGTGGCCTT	GGTCGCTGAGCACATAATCCA	[64]
ASCT-2	NM_001145144.2	TCCTCTTCACCCGCAAAAACC	CCACGCCATTATTCTCCTCCAC	[65]
Syn-2	NM_207582.3	GCCTGCAAATAGTCTTCTTT	ATAGGGGCTATTCCCATTAG	[66]
MFSD2A	NM_001136493.3	CTCCTGGCCATCATGCTCTC	GGCCACCAAGATGAGAAA	[67]
IL-1β	NM_000576.2	CCACCTCCAGGGACAGGATA	AACACGCAGGACAGGTACAG	[60]
IL-6	NM_000600.3	TGCAATAACCACCCCTGACC	ATTTGCCGAAGAGCCCTCAG	[60]
IL-10	NM_000572.2	ACATCAAGGCGCATGTGAAC	CACGGCCTTGCTCTTGTTTT	[60]
TNF-α	NM_000594.3	CCCGAGTGACAAGCCTGTAG	TGAGGTACAGGCCCTCTGAT	[60]
MCP-1	NM_002982.4	AGAATCACCAGCAGCAAGTGTCC	TCCTGAACCCACTTCTGCTTGG	[18]
INF-γ	NM_000619.2	TCAGCTCTGCATCGTTTTGG	GTTCCATTATCCGCTACATCTGAA	[60]
TLR-3	NM_003265.3	GCGCTAAAAAGTGAAGAACTGGAT	GCTGGACATTGTTCAGAAAGAGG	[64]
TLR-4	NM_003266.4	CCCTGAGGCATTTAGGCAGCTA	AGGTAGAGAGGTGGCTTAGGCT	[64]
TLR-9	NM_017442	TGC CCA AAC TGG AAG TCC TC	TAA GGT TGA GCT CTC GCA GC	[64]
GAPDH	NM_001256799.3	GTCTCCTCTGACTTCAACAGCG	ACCACCCTGTTGCTGTAGCCAA	[64]

## Data Availability

The data are available at the corresponding authors.

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
