# Peer review of "Preliminary Evidence of the Differential Expression of Human Endogenous Retroviruses in Kawasaki Disease and SARS-CoV-2-Associated Multisystem Inflammatory Syndrome in Children"

_ijms, 2023, doi:10.3390/ijms242015086_

Round 1
Author Response
Preliminary evidence of the differential expression of Human 2 endogenous retroviruses in Kawasaki disease and SARS-CoV- 2-associated multisystem inflammatory syndrome.
This pilot study aims to analyze the expression of HERVs, HERV-related genes and immune mediators in children during acute and subacute phases of Kawasaki Disease (KD) and multisystem inflammatory syndrome in children (MIS-C) compared with COVID pediatric patients and healthy controls, to identify a possible specific disease pattern of expression and its correlation con the common inflammatory markers for each clinical feature.
This work is interesting, methodologically correct, and well supported by literature.
We are glad the reviewer appreciated our work and we would like to thank for the thoughtful review of the manuscript.
My concerns are detailed below.
Title:
I would add “SARS-CoV- 2-associated multisystem inflammatory syndrome in children”.
We modified the title and throughout the full text manuscript accordingly.
Abstract:
The abstract adequately summarizes the content of the article.
Line 30: please spell it out CRP and remove “a marker of inflammation”.
We modified this point and throughout the full text manuscript accordingly.
Introduction:
The introduction is appropriate, but too long.
- I would shorten the paragraph, shortening the first epidemiological part and giving more importance to the second part explaining the possible HERV role in MIS-C and KD pathogenesis. I would clarify your thesis about the theoretical role of those gene expressions in the pathogenesis of diseases: they can trigger inflammation? Are proteins considered exogenous? Have proteins a direct pro-inflammatory function? Is it just a marker of the disease? I also would explain in a very summarized way the function of the studied protein (sow lines 244-246).
We shortened the first epidemiological part as suggested and we added new concerns about the potential role of HERVs in this setting including the interplay with the immune system and peculiar function of the different studied HERV protein.
- Line 58 “MIS-C” you miss “in children” in the acronym.
We modified as indicated and throughout the full text manuscript accordingly.
- At the end of the paragraph please add the secondary aim: to evaluate correlation between HERVs and common inflammatory markers.
We added the secondary aim as suggested in the revised version of the manuscript.
Methods:
It is quite repetitive in some parts and could be improved in its organization.
- I would call the first paragraph “study design and participants” instead of “Patients and Healthy Controls”.
We modified as indicated.
- Please, in the first paragraph present the whole study population and controls, with the sample size.
We modified as indicated.
- Please write a different methods paragraph called “ethical approval” between the first one and “samples preparation”, where you summarize all about study consent signature and ethical approval (lines 98-102 and 138-140).
We added the new paragraph called “ethical approval”.
- I would a paragraph “data collection” to put the information in lines 120-131
We added the new paragraph “data collection”.
Results:
The results are interesting and clear, but this part is quite long. I suggest you mention the most important findings in the test and leave the images to explain the rest.
- Line 226: please remove “p-values 226 for all the comparisons are detailed in panel D”, because you already wrote it.
We removed any repetitions throughout the results paragraph.
- Line 244-246 spells out the acronym when you cited it at the first time in the test.
We corrected as indicated.
- Is there any significant movement of HERVs in responders and non-responders' patients? Why did you choose to non-mention it in the test?
We thank the reviewer, due the low number we avoid to stratified patients as responder and non-responder
Discussion:
This section is well exposed, but it could be better explaining the hypothetical role of HERV in the syndromes: it has pro-inflammatory function and, in some cases also anti-inflammatory. Underline better your point of view about the topic.
- Please add the difference between male/female ratio in your population as for study limitation.
We added this limitation in the revised version of the manuscript.
- I would also underline the fact that this study only allows you to prove the correlation between the studied parameters more than a causative relation.
We completely agree with the reviewer. The present results allow only to point out correlations that are not absolutely related to cause-and-effect events. We modified the text to avoid providing misleading findings.
- Moreover, as study limitation, I would mention the small sample size mostly for severe deceased patients and for the immunological study.
We added this limitation in the revised version of the manuscript.
- Are there any the perspectives of the HERV pathogenesis in the cure of MIS-C? If there are, please write it in the test.
We are far to suggest the potential use of HERVs as potential therapeutic targets in this setting considering that our work is a pilot study providing only preliminary evidence that need to be corroborated by other studies with a larger sample size.

Reviewer 2 Report
1. MIS-C clinical features overlap with KD. Spike of SARS-CoV-2 trigger HERV activation and HERV is contributing factor in COVID-19. How about clinical features of SARS-COV2 and HERV?
2. HERV-W positively correlated with CRP. Is it correlate with HERV-S with CRP?
3. MFSDA2A primer is constructed from which gene?
4. Are there any significant differences of cytokines between HC, COV, KD, MIS-C between acute and subacute phase samples? How to identify acute and subacute samples after onset of fever?
5. Figure 1D and 2D tables are difficult to see. We cannot see clearly on significant value. Please revise the table.
6. Which Cytokines are downregulation or upregulation sharing among COV, KD, HERV?
7. In figure 1 and 2, please add Y axis . what is 1, 10, 1000?
Author Response
Comments and Suggestions for Authors
- MIS-C clinical features overlap with KD. Spike of SARS-CoV-2 trigger HERV activation and HERV is contributing factor in COVID-19. How about clinical features of SARS-COV2 and HERV?
We thank the reviewer for this comment. As stated, studying the potential role of HERVs in COVID-19, our research group and others demonstrated that HERVs are highly expressed in the blood cells and at the initial site of infection in patients, closely related to disease severity. In the present study, likely due to the small sample size, the deregulation of HERV expression was not associated to clinical parameters exception made for the correlation between HERV-W and CRP levels in MIS-C during the acute phase.
- HERV-W positively correlated with CRP. Is it correlate with HERV-S with CRP?
In the present study, the deregulation of HERV expression was not associated to clinical parameters exception made for the positive correlation between HERV-W and CRP levels in MIS-C during the acute phase. This observation is in line with a previous study in which it has been demonstrated that HERV-W GAG or ENV antigenemia positively correlated with CRP levels in schizophrenic patients, corroborating the hypothesis that HERV-W sustain a chronic inflammatory state. This point has been clarified in the discussion paragraph in the revised version of the manuscript
- MFSDA2A primer is constructed from which gene?
The primer pairs for MFSD2A derived from the paper published by Toufaily et al (https://doi.org/10.1016/j.placenta.2012.10.012) – reference n. 43 - and amplified the variant 1 representing the longest transcript encoding the longest isoform. The primer pairs amplified a sequence belonging to the CoDing sequence.
- Are there any significant differences of cytokines between HC, COV, KD, MIS-C between acute and subacute phase samples?
In our cohort study, the acute and the subacute phases are available only for KD and MIS-C. Concerning the differences in terms of cytokines related to the progression of the disease, in KD higher levels of IL-10, TNF- a, and MCP-1 were found during acute phase compared with the subacute phase while in MIS-C, higher levels of TNF-a, TLR-3, TLR-4, TLR-7, and TLR-9 were found during acute with respect to subacute phase, while IL-6 and IL-10 levels were found to be higher in subacute phase. We added this specific point in the discussion section to better emphasize our findings.
How to identify acute and subacute samples after onset of fever? The definition of acute and subacute stage is reported in the section “methods” in the revised version of the manuscript, so it’s the timing of sample collection. The samples collected during acute stage of the disease were considered of the acute stage, and those collected during the subacute stage were considered of the subacute stage of the disease. Patients with COVID-19 were studied only during the acute stage.
- Figure 1D and 2D tables are difficult to see. We cannot see clearly on significant value. Please revise the table.
We revised the tables to optimize the resolution.
- Which Cytokines are downregulation or upregulation sharing among COV, KD, HERV?
We thank the reviewer for raise this point. We added a more detailed description in the discussion section in the revised version of the manuscript.
- In figure 1 and 2, please add Y axis . what is 1, 10, 1000?
We apologize we forgot to insert Y axis which identify the relative expression of the target gene with respect to the housekeeping gene in logarithmic scale. We uploaded a new version of the figure.

Reviewer 3 Report
This manuscript compares HERV expression in pediatric patients with Kawasaki Syndrome and MIS-C.
The study is well designed and provides very intersting results showing that HERVs can be activated in these syndromes in post-infectious settings and may drive hyper-inflammatory immune responses.
The study unveils important features:
-HERVs can be upregulated and/or activated in these two syndromes. One of them has a readily identified viral trigger (SARS-CoV-2 in MIS-C) and viruses from recent infection preceding the Kawasaki syndrome may also have played a similar role. Interestingly the transcription of HERV-W and HERV-K are significantly elevated in both acute or subacute syndromes, while some physiologically modified copies from the HERV-W family (Syncytin-1) are co-regulated along with the corresponding membrane receptors in the same conditions. This calls for further studies at the protein level but paves the way for further targeted analyses of expected good value.
-There is a differential pattern between transcriptional activation of some HERV genes between MIS-c and Kawasaki syndrome, which may be used as discriminating biomarkers in relevant studies: Syn-2 and MFSD2A (gene locus containing an HERV insertion) show elevated RNA levels in MIS-C only.
-IL-6 transcription correlates with both HERV-W and HERV-K RNA levels, but IL-10 shows a positive correlation with HERV-K RNA levels and an inverse correlation with HERV-Wlevels. IL-10 being mostly involved in immunoregulation and in anti-inflammatory responses, it may thereby differentiate the effects of HERV-K versus HERV-W in a dominant pro-inflammatory context (IL-6 dominating in both cases and IL-10 being an antagonistic anti-inflammatory response of moderate efficacy in this context, or ultimately only).
-surprisingly, TLRs RNA are not elevated but in MIS-C.
These main results may be better highlighted and, most of all, it is needed to specify that the whole study is relying upon RNA quantification (transcriptional activity), which is made ambiguous when writing about end-protein or gene “expression.
This terminology should be made clear from the title and the abstract, to make this excellent study all the more accurate.
only typos or small mistakes
Author Response
This manuscript compares HERV expression in pediatric patients with Kawasaki Syndrome and MIS-C.
The study is well designed and provides very intersting results showing that HERVs can be activated in these syndromes in post-infectious settings and may drive hyper-inflammatory immune responses.
We are glad the reviewer appreciated our work and we would like to thank for the thoughtful review of the manuscript.
The study unveils important features:
-HERVs can be upregulated and/or activated in these two syndromes. One of them has a readily identified viral trigger (SARS-CoV-2 in MIS-C) and viruses from recent infection preceding the Kawasaki syndrome may also have played a similar role. Interestingly the transcription of HERV-W and HERV-K are significantly elevated in both acute or subacute syndromes, while some physiologically modified copies from the HERV-W family (Syncytin-1) are co-regulated along with the corresponding membrane receptors in the same conditions. This calls for further studies at the protein level but paves the way for further targeted analyses of expected good value.
We thank the reviewer for emphasizing these crucial findings and we completely agree with the need to move toward new studies focused at the protein level. We added this point in the discussion section of the revised version of the manuscript.
-There is a differential pattern between transcriptional activation of some HERV genes between MIS-c and Kawasaki syndrome, which may be used as discriminating biomarkers in relevant studies: Syn-2 and MFSD2A (gene locus containing an HERV insertion) show elevated RNA levels in MIS-C only.
We agree with the reviewer. Only patients with MIS-C, in addition to having high values of HERVs, Syn-1 and its putative receptors, showed elevated expression levels of Syn-2 and its receptor named MFSD2A. This expression profile is also maintained in MIS-C patients during the subacute phase towards the resolution of the disease. Thus, as suggested in the discussion paragraph, Syn-2 and MFSD2A could be suggested as hallmark of MIS-C, able to characterize the disorder from others with overlapping symptomatology.
-IL-6 transcription correlates with both HERV-W and HERV-K RNA levels, but IL-10 shows a positive correlation with HERV-K RNA levels and an inverse correlation with HERV-Wlevels. IL-10 being mostly involved in immunoregulation and in anti-inflammatory responses, it may thereby differentiate the effects of HERV-K versus HERV-W in a dominant pro-inflammatory context (IL-6 dominating in both cases and IL-10 being an antagonistic anti-inflammatory response of moderate efficacy in this context, or ultimately only).
We agree with the reviewer about this point. As such, the positive correlations among HERVs and IL-6 are in line with the findings indicating that HERVs are able to sustain the pro-inflammatory state in peculiar clinical setting (i.e. COVID-19) and on the other hand, the correlations among HERVs and IL-10 corroborated the immunosuppressive activity as intrinsic property of some HERVs.
-surprisingly, TLRs RNA are not elevated but in MIS-C.
Concerning COVID-19 and MIS-C patients, a general increase of all TLRs evaluated have been found and this was in agreement with the well-known functions of viral elements and damage associated host molecules able to act as TLR ligands in these diseases.
These main results may be better highlighted and, most of all, it is needed to specify that the whole study is relying upon RNA quantification (transcriptional activity), which is made ambiguous when writing about end-protein or gene “expression. This terminology should be made clear from the title and the abstract, to make this excellent study all the more accurate.
We apologize for the inaccuracy and we carefully reviewed the full manuscript to avoid misleading
Comments on the Quality of English Language
only typos or small mistakes
We carefully reviewed the full text manuscript for typos and other mistakes.
